# Seroprevalence of three paramyxoviruses; Hendra virus, Tioman virus, Cedar virus and a rhabdovirus, Australian bat lyssavirus, in a range expanding fruit bat, the Grey-headed flying fox (*Pteropus poliocephalus*)

Wayne S. J. Boardman[1]*, Michelle L. Baker[2], Victoria Boyd[2], Gary Crameri[2], Grantley R. Peck[3], Terry Reardon[4], Ian G. Smith[1,5], Charles G. B. Caraguel[1ᵒ], Thomas A. A. Prowse[6ᵒ]

1 School of Animal and Veterinary Sciences, University of Adelaide, Adelaide, South Australia, Australia, 2 CSIRO Health and Biosecurity Business Unit, Australia Animal Health Laboratory, Geelong, Victoria, Australia, 3 CSIRO, Australian Animal Health Laboratory, Geelong, Victoria, Australia, 4 South Australia Museum, Adelaide, South Australia, Australia, 5 Zoos South Australia, Adelaide, South Australia, Australia, 6 School of Mathematical Sciences, University of Adelaide, Adelaide, South Australia, Australia

ᵒ These authors contributed equally to this work.
* wayne.boardman@adelaide.edu.au

## Abstract

Habitat-mediated global change is driving shifts in species' distributions which can alter the spatial risks associated with emerging zoonotic pathogens. Many emerging infectious pathogens are transmitted by highly mobile species, including bats, which can act as spill-over hosts for pathogenic viruses. Over three years, we investigated the seroepidemiology of paramyxoviruses and Australian bat lyssavirus in a range-expanding fruit bat, the Grey-headed flying fox (*Pteropus poliocephalus*), in a new camp in Adelaide, South Australia. Over six, biannual, sampling sessions, we quantified median florescent intensity (MFI) antibody levels for four viruses for a total of 297 individual bats using a multiplex Luminex binding assay. Where appropriate, florescence thresholds were determined using finite mixture modelling to classify bats' serological status. Overall, apparent seroprevalence of antibodies directed at Hendra, Cedar and Tioman virus antigens was 43.2%, 26.6% and 95.7%, respectively. We used hurdle models to explore correlates of seropositivity and antibody levels when seropositive. Increased body condition was significantly associated with Hendra seropositivity (Odds ratio = 3.67; p = 0.002) and Hendra virus levels were significantly higher in pregnant females (p = 0.002). While most bats were seropositive for Tioman virus, antibody levels for this virus were significantly higher in adults (p < 0.001). Unexpectedly, all sera were negative for Australian bat lyssavirus. Temporal variation in antibody levels suggests that antibodies to Hendra virus and Tioman virus may wax and wane on a seasonal basis. These findings suggest a common exposure to Hendra virus and other paramyxoviruses in this flying fox camp in South Australia.

**Data Availability Statement:** All relevant data are within the manuscript and its Supporting Information files.

**Funding:** The author(s) received no specific funding for this work.

**Competing interests:** The authors have declared that no competing interests exist.

## Introduction

The emergence of zoonoses from wildlife represents an increasingly significant threat to global public health [1]. Bats (Order Chiroptera) are the reservoir host of several significant groups of emerging zoonotic viruses including the paramyxoviruses, (e.g. Hendra virus and Nipah virus), coronaviruses, filoviruses and lyssaviruses [2–5]. In Australia, spill-over of three viruses associated with bats of the genus *Pteropus*, also known as flying foxes, has led to morbidity and mortality in domestic animals and humans. They include two paramyxoviruses, Hendra virus and Menangle virus, and a rhabdovirus, Australian bat lyssavirus [6–10]. Research into the ecology of these viruses led subsequently to the discovery of several new paramyxoviruses, including Cedar virus, Hervey virus, Yeppoon virus, Grove virus, Teviot virus [11] and Tioman virus [12]. Tioman virus, closely related to Menangle virus [8] is the only one of these viruses to be associated with disease. It has been associated with sub-clinical infection in humans and still births and fetal abnormalities in pigs [13].

The Grey-headed flying fox (*Pteropus poliocephalus*), one of four species of flying foxes found on mainland Australia, is classified nationally as vulnerable under the *Environment Protection and Biodiversity Conservation Act 1999* [14]. The geographical distribution and migration of Grey-headed flying foxes and other pteropodids is dictated by the distribution and phenology of food plants. These bats regularly move long distances in search of ephemeral floral and fruit resources in native forests [15,16]. Aggregations of flying foxes can increase rapidly during highly productive flowering events [17]. Recently, Grey-headed flying foxes were distributed from Ingham in Queensland along the coastal belt of eastern Australia to Melbourne in Victoria. As natural food resources have declined coincident with substantial (*c.* 75%) loss of native forest throughout the south-eastern coastal areas of Australia, Grey-headed flying foxes have sought alternative food sources, sometimes forming new colonies in urban landscapes [18,19].

Habitat loss and fragmentation reduce not only the quantity of food available to wildlife, but also the connectivity of foraging patches, particularly if seasonally important resources have been removed [16]. In contrast, anthropogenic resource subsidies, which favour monoculture (e.g., fruit orchards) and introduced species, change the composition and seasonality of available food and the overall nutritional landscape [16]. Recently, Grey-headed flying foxes formed camps in Canberra and western parts of Victoria and, during the 2010 winter, approximately 1300 individuals migrated to Adelaide, South Australia, thereby expanding the former range of the species. Since that time, the population in Adelaide's Botanic Park, which is a popular recreational location, has increased to approximately 20,000 individuals due to births and continued immigration(November 2019), despite seasonal emigration and substantial bat mortality events during extreme heat waves in summer. Concerns have been raised that the bat camp may constitute a biohazard to the public and to domestic animals. Indeed, Australian bat lyssavirus was detected in a Grey-headed flying fox from the camp in 2012 [20]. Since then, another twenty-six Grey-headed flying foxes from the camp were opportunistically tested for the virus of which none tested positive. However, uncertainty remains about the endemicity of Australian bat lyssavirus in the Adelaide camp.

When investigating the infection dynamics of emerging viruses in bat colonies, direct viral detection and identification is important but is technically limited due to restricted distribution of the virus in organs and transient viral shedding in biological fluids. Complementing virus detection, the exposure to specific viruses can be measured by detecting antibodies against those viruses in bat sera. Antibodies are generally present for months or even years even if the virus is scarely distributed or even after it is cleared from the animal. As a result, viral seroprevalence monitoring has often been the first line of investigation for emerging bat

zoonoses [21–25]. However, interpreting serological results is challenging [26] in part due to variation in the magnitude and longevity of antibody responses to different viruses, and the time of collection of serum post infection [27]. Furthermore, antibodies may cross-react with or cross-neutralize related viral antigens, which can limit the specificity of assays.

Serum viral neutralization tests (SNTs) have been considered the reference method for detecting specific antibodies to Hendra virus [28]. However, the use of SNTs is logistically constraining because the highest level of biocontainment (Biosafety level 4) is required to maintain the live viral cultures used for the neutralization assays. Instead, IgG enzyme-linked immuno-sorbant assays (ELISAs) and Luminex based assays [29] have been favoured because they can be performed under standard biosafety conditions [30]. Luminex based fluorescent micro-sphere binding assays [29] are a sensitive method for detection and quantification of antibod-ies against Hendra and Nipah viruses [15, 22, 31, 32] and Australian bat lyssavirus [33] in bat sera. The target antigen for Hendra virus and Cedar virus is recombinant soluble G protein [32] while the target antigens for Tioman virus and Australian bat lyssavirus are nucleopro-teins. Luminex assays have been used internationally to detect henipavirus antibodies in bats and other species; including West African fruit bats and domestic pigs [22,34–36], pteropodid bats in Papua New Guinea [15] and *Pteropus vampyrus* bats in Indonesia [37].

Serological evidence of infection with Hendra virus has been shown in all four species of pteropodid bat that occur on mainland Australia, throughout their respective ranges [21, 38]. There is evidence to suggest that two species, namely the Black flying fox (*Pteropus alecto*) and the Spectacled flying fox (*P. conspicillatus*), play the most active role in the transmission of Hendra virus to horses [38]. Hendra virus is shed in the urine, an important vehicle for trans-mission in Black flying foxes [10, 38–39], and the virus has been detected in Grey-headed fly-ing fox uterine fluid which provides evidence for possible transmission at birthing period which lasts from late September to early December [40] in this species [41, 42].

Here, we surveyed the exposure of Adelaide's recently established Grey-headed flying fox population to protein antigens of Hendra virus, Cedar virus, Tioman virus and Australian bat lyssavirus over a three year period. We used results from Luminex antibody binding assays to develop a finite-mixture model to identify thresholds for defining seropositive flying foxes to characterise seroprevalence for these four viruses. Next, we used a negative-binomial hurdle model and investigated individual-level correlates of (i) seropositivity and (ii) antibody level following seroconversion. We hypothesised that Hendra virus seroprevalence and antibody levels would be associated with reproductive status as previously reported [21,23] and that Australian bat lyssavirus seroprevalence would be apparent given the prior finding of an indi-vidual carrying the virus in 2012 [20].

## Materials and methods

We followed the Consortium for the Standardization of Influenza Seroepidemiology (CON-SISE) guidelines [43] for the reporting of seroepidemiologic studies. Animal Ethics approval was obtained from The University of Adelaide (S-2015-028) and a wildlife scientific permit from the SA Department of Environment and Water (M-23671-1,2 and 3) prior to commence-ment of this project.

### Sampling

**Study population.** The target and source population were the grey-headed flying foxes from the only known camp in SA (Fig 1) and established in Adelaide's Botanic Park [approxi-mate GPS coordinates: 34˚54'56 S, 138˚36'24 E]. The camp was sampled over six surveys at approximately six-month intervals between August 2015 (winter) and February 2018

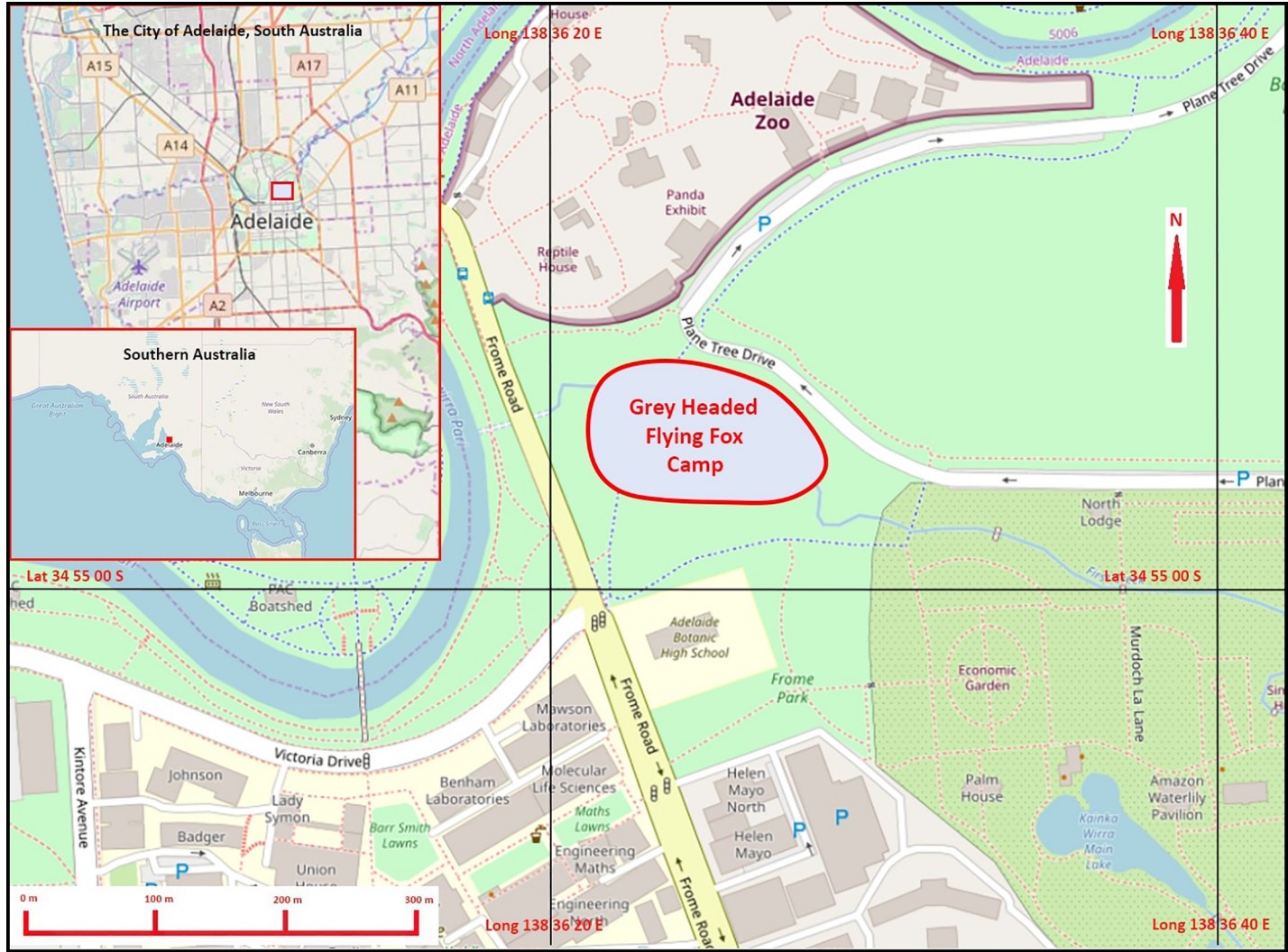

**Fig 1. Location of the Grey-headed flying fox camp in Adelaide's Botanic Park and relationship to central Adelaide and Southern Australia.**

(summer), with the aim of trapping > 50 animals per survey, which constituted our study population.

Location and extent (red line) of the Grey-headed flying fox camp in Adelaide's Botanic park, showing proximity to Adelaide Zoo where bats were processed. Insets illustrate central Adelaide and Southern Australia to show geographical relationships. Geodata from Open-StreetMap was downloaded via the Maperitive application and the map was rendered with further information supplied by the author.

**Bat handling, serum and data collection.** Study animals were captured at the roost site using 12 or 18 m long mist nets (Ecotone, Gdynia, Poland) installed beneath the camp. Mist nets were raised 20 m above the ground before bats returned from their nightly foraging activity. As each bat became entrapped, the net was lowered, the bat handled with care using thick leather gloves to assure handlers' health and safety, then transferred into pillow cases and relocated to the Animal Health Department of the adjacent Adelaide Zoo. The net was then elevated to 20 m above the ground to catch further bats. This continued until all bats had returned to the camp. Isoflurane (Isoflurane, Laser Animal Health) was used to anaesthetise

bats during data and sample collection following the protocol described by Jonsson et al, 2004 [44]. Each bat was permanently identified using a passive integrated transponder tag (Trovan, Microchips Australia Pty, Keysborough, Victoria) inserted subcutaneously between the scapulae. A small amount of fur was clipped from the chest to rapidly identify recaptures at a given survey. In order to prevent dehydration during their short term confinement, 20–40 mL Hartmann's fluid were injected subcutaneously between the scapulae. Approximately 3–4 mL of blood was collected via venepuncture of the propatagial or brachial vein into 4 mL serum tubes using 22-gauge needles and 3–5 mL syringes. These were allowed to clot overnight at room temperature and then at 4°C before centrifugation (5,000 rpm for 5 minutes) and separation of serum, which was subsequently stored at -80°C. After sampling, bats were placed into pillowcases to fully recover from the anesthesia before release into the camp.

For each bat, we recorded: (i) sex, (ii) body weight (BW; g), (iii) body condition score (scale of 1 to 5 based on physical palpation of the pectoral musculature by the same person), (iv) forearm length (FAL; mm); elbow to wrist length using vernier callipers, (v) estimated age as described by Hall and Richards [40] (including teeth wear, nipple size for females and enlarged penis/testes for males), (vi) reproductive status (for females; pregnant vs not pregnant by abdominal palpation, lactating vs non lactating by expression of milk and for males; enlarged penis/testes vs small penis/testes). For an objective estimate of the body condition, we also derived a body condition index (BCI) for each individual, calculated subsequently as BCI = $1{,}000^{*}(BW/FAL^2)$.

## Serology for Hendra virus, Cedar virus, Tioman virus and Rabies virus

Serum samples experienced two freeze/thaw cycles prior to testing. Antibodies against Hendra virus, Cedar virus, Tioman virus, and Australian bat lyssavirus antigens were detected at the Australian Animal Health Laboratory in Geelong, Victoria using multiplex microsphere assays (Luminex, Austin, USA) as described previously [29]. The conformational status of the viruses used were the following; soluble native-like oligomeric G envelope glycoproteins of HeV and CedV ($sG_{tet}$) were produced from stable expressing FreeStyle[TM] 293F cell lines [45,46], Tioman virus was a nucleocapsid protein expressed in the yeast *Saccharomyces cerevisiae* [47], and Australian bat lyssavirus was a nucleocapsid protein prepared in E.coli [48]. Briefly, prior to analysis, serum samples were first heat treated at 56°C for 30 minutes to inactivate complement then the assay proteins were coupled to individual microsphere bead sets, of predetermined numbers of magnetic beads, MagPlex[®] (Luminex, Northbrook, USA). These were added to each well and then mixed with bat sera at a dilution of 1:50. The bound antibody was detected using biotinylated Protein A (Pierce, Rockford, USA) together with biotinylated Protein G (Pierce, Rockford, USA) followed by streptavidin–phycoerythrin (Qiagen Pty Ltd, Australia). The assay was read using a Bio-Plex Protein 200 Array System integrated with Bio-Plex Manager Software (v 6.2) (Bio Rad Laboratories, CA, USA) calibrated on the high setting. Each sample was tested in a well with thousands of beads and the florescence results of 100 beads were recorded as the median florescent intensity (MFI) that excludes outliers and are correlated with antibody concentration. Positivity thresholds for the Luminex serological assay have not been defined for Australian flying foxes due to the lack of negative and single-infection control serum [32] and were therefore estimated using finite mixture modelling (see below).

## Statistical analysis

**Demographic analysis.** Two-sample t-test statistics were used to identify any differences in BW, FAL and BCI across demographic classes (sub-adult males and females and adult males and adult pregnant and non pregnant females) and between winter and summer.

**Estimating MFI thresholds for classifying seropositive animals.** MFI values were log-transformed prior to analysis to approximate a normal distribution and enable parametric analyses. We used finite mixture modelling in the statistical package Stata v15.1 (College Station, Texas, USA) to identify the presence of more than one sub-population under the assumption of normal distribution. Models assuming up to three mixed distributions were run and their parsimony compared using Akaike Information Criterion (AIC) and Bayesian Information Criterion (BIC). The model with the lowest AIC and/or BIC was selected as final. When the single distribution model fitted best, the distribution was assumed to be the non-seroconverted bats if MFI values were in the lower end of the range and seroconverted bats if in the higher end of the range. When the two distributions model fitted best, the distributions were considered as the non-seroconverted and the seroconverted bats, respectively, according to their values' range. When the three distributions model fitted best, the distribution with the lowest value range was considered as the non-seroconverted bats and the other two distributions (with higher value ranges) as two sub-groups of seroconverted bats. When two distributions fully overlapped, these were considered as one single distribution because the readings had no discriminative ability. For best fit models with more than one mixed distribution that overlapped partially, threshold values were determined visually at the MFI value for which two predicted normal distributions intersected. These threshold values were used to classify (imperfectly) individual bats as 'seronegative' or 'seropositive' (or 'intermediate positive' when three distributions were identified).

**Hurdle modelling of seropositivity and antibody levels.** To investigate correlates of seropositivity and MFI levels, we used a hurdle regression model which included two components; (i) the 'hurdle' component, which modelled the probability of being seropositive (as defined using the estimated lower threshold value); and (ii) a negative binomial count component, which modelled the antibody value (expressed as MFI) conditional on seropositivity. Explanatory variables investigated were those measured or observed during the trapping i.e. sex (male or female), age class (subadult < 2.5 years or adult ≥ 2.5 years), body weight (g), forearm length (mm), catching session (1–6), season (winter or summer), pregnancy status, lactation status, and body condition index (BCI). However, we excluded the effect of season from the final model due to strong collinearity between season and time of survey. Hurdle models were implemented within the *R* software for statistical computing (version 3.2.3) using the package *pscl*. The extent of co-seropositivity for all four viruses was also assessed using the negative binomial hurdle regression model investigating the same explanatory variables.

**Demonstration of zero seroprevalence for Australian bat lyssavirus.** The probability that the Adelaide camp is free from Australian bat lyssavirus was estimated using the historical survey analysis outlined by Cameron (2014) [49]. The probability of freedom from Australian bat lyssavirus was uncertain before the first survey and a prior value of 50% was used. In the absence of published information on the diagnostic accuracy of the multiplex Luminex assay, we optimistically assumed that this method had perfect accuracy for Australian bat lyssavirus antibodies. Similarly, little information is available on differential risks of Australian bat lyssavirus exposure across bat demographics. Therefore, the risk of Australian bat lyssavirus 'exposure' was assumed constant across the bat camp strata (i.e. risk independent modelling). The 'open' nature of our study population was taken into account by including a 'between-survey' probability of exposure from and/or introduction of immigrating exposed bats into the model. Freedom from Australian bat lyssavirus seroprevalence was deemed achieved if the estimated probability of freedom was ≥ 95%.

## Results

### Demographic information

A total of 301 Grey-headed flying foxes were captured over six surveys. Four individual flying foxes were recaptured once each during this period. Demographics including sex, age, weight, forearm length and BCI varied across seasons reflecting the seasonality of the species' reproduction and feeding opportunities (Tables 1 and 2). Approximately, two thirds were females and two thirds were adults. The overall percentage of adults was similar between sexes (61.7% vs 69.9%). Among the adult females, 31.9% (37/116) were pregnant at capture. Sex and age-class representation was similar across sampling and seasons. Overall the BCI was higher in winter (mean = 28.2, range = 20.2–36.1) than summer (mean = 25.3, range = 13.8–35.8) (+ 2.9 units, $p < 0.001$) mainly driven by the sub-adult BCI being higher in winter (n = 52, mean = 25.8, range = 20.2–29.6) than summer (n = 52, mean = 21.1, range = 13.8–29.2) (+4.7 units, $p < 0.001$).

### Serology thresholds and serological prevalence

Multiplex serology was conducted on 301 serum samples (comprising 297 individual bats, with 4 recaptures). With the exception of Australian bat lyssavirus, these assays yielded multi-modal distibutions for log-transformed MFI (Fig 2). Three mixed distributions were identified for Hendra virus, Cedar virus and Tioman virus, and two cut-offs, a lower and upper, were determined visually (Fig 2). Upper and lower thresholds for the Hendra virus serology were determined as the natural antilogarithms of 5.85 and 8.67, respectively (MFI 347 and 5825, respectively). Upper and lower thresholds for the Cedar virus serology were determined as the natural antilogarithms of 5.76 and 7.44, respectively (MFI 317 and 1702, respectively). Upper and lower thresholds for the Tioman virus serology were determined as the natural antilogorithms of 6.37 and 7.38 respectively (MFI 584 and 1603, respectively). A single distribution of assumed non-seroconverted animals was identified for Australian bat lyssavirus. Using the lower threshold values, 26.6% of the bats were seropositive for Cedar virus, 43.2% of the bats for Hendra virus and 95.7% of the bats for Tioman virus were seropositive (Table 3).

**Table 1. Summary demographic statistics–all captures.**

| Demographic Classes | BW (g) | | | FAL (mm) | | | BCI | | |
|---|---|---|---|---|---|---|---|---|---|
| | n | Mean (SD) | Range | n | Mean (SD) | Range | n | Mean (SD) | Range |
| Sub-adult (all) | 106 | 550 (115.4) | 266–772 | 104 | 153.0 (7.3) | 127.0–171.0 | 104 | 23.5 (3.6) | 13.8–29.6 |
| Sub-adult Females | 72 | 566 (110.8) | 291–739 | 70 | 153.8 (6.8) | 140.0–170.5 | 70 | 23.9 (3.6) | 13.8–29.6 |
| Sub-adult Males | 34 | 517 (119.5) | 266–772 | 34 | 151.7 (7.8) | 127.3–162.6 | 34 | 22.6 (3.4) | 16.4–29.2 |
| Adult (all) | 195 | 764 (90.0) | 563–1,008 | 193 | 164.0 (4.8) | 152.0–176.8 | 104 | 23.5 (3.6) | 13.8–29.6 |
| Adult Females | 116 | 743 (78.7) | 563–1,005 | 114 | 163.0 (4.7) | 152.0–174.4 | 114 | 27.9 (2.8) | 22.0–35.1 |
| Pregnant | 37 | 792 (91.6) | 600–1005 | 37 | 162.3 (5.0) | 152.0–171.4 | 37 | 30.0 (2.7) | 22.8–34.9 |
| Not Pregnant | 79 | 720 (60.0) | 563–963 | 79 | 164.0 (4.6) | 154.0–173.0 | 79 | 27.0 (2.3) | 22.0–35.2 |
| Adult Males | 79 | 794 (96.0) | 585–1,008 | 79 | 165.2 (4.7) | 155.1–176.8 | 34 | 29.1 (2.8) | 22.6–36.1 |
| All Females | 188 | 676 (126.1) | 291–1,005 | 184 | 159.5 (7.2) | 140.0–173.4 | 184 | 26.4(3.7) | 13.8–35.2 |
| All Males | 113 | 711 (164.2) | 266–1,008 | 113 | 160.6 (9.0) | 127.3–176.8 | 113 | 27.1 (4.2) | 16.4–36.1 |
| Total | 301 | 689 (142.4) | 266–1,008 | 297 | 160.0 (8.0) | 127.3–176.8 | 297 | 26.7 (3.9) | 13.8–36.1 |

Summary demography statistics (number (n), mean, standard deviation (SD), range) of 301 Grey-headed flying foxes captured over six surveys between September 2015 and February 2018 from the Adelaide Camp, Botanic Park, Adelaide. BW = Body weight, FAL = forearm length, BCI = body condition index = $1,000*(BW/FAL^2)$

**Table 2. Summary demographic statistics–winter versus summer.**

| Demographic Classes | WINTER | | | | | | | | | SUMMER | | | | | | | | |
|---|---|---|---|---|---|---|---|---|---|---|---|---|---|---|---|---|---|---|
| | BW (g) | | | FAL (mm) | | | BCI | | | BW (g) | | | FAL (mm) | | | BCI | | |
| | n | Mean (SD) | Range | n | Mean (SD) | Range | n | Mean (SD) | Range | n | Mean (SD) | Range | n | Mean (SD) | Range | n | Mean (SD) | Range |
| Sub-adult | 53 | 628.6 (42.8) | 516–695 | 52 | 156.3 (4.1) | 146.0–166.0 | 52 | 25.8 (2.0) | 20.2–29.6 | 53 | 472 (112.0) | 266–772 | 52 | 149 (7.8) | 127.0–161.0 | 52 | 21.1 (3.3) | 13.8–29.2 |
| Females | 40 | 629 (44.5) | 516–695 | 39 | 156.2 (4.2) | 147.0–166.0 | 39 | 25.9 (2.1) | 20.2–29.6 | 32 | 486 (118.0) | 291–739 | 31 | 150.5 (8.0) | 140.0–170.5 | 31 | 21.3 (3.5) | 13.8–29.0 |
| Males | 13 | 626 (38.6) | 559–675 | 13 | 156.4 (3.8) | 147.8–162.0 | 13 | 25.6 (1.6) | 22.7–28.4 | 21 | 449 (100.7) | 266–772 | 21 | 146.1 (7.0) | 127.3–162.6 | 21 | 20.8 (2.8) | 16.4–29.2 |
| Adult | 90 | 790 (99.2) | 585–1008 | 90 | 163.3 (5.0) | 152.0–173.0 | 90 | 29.5 (2.9) | 22.6–36.1 | 105 | 741 (77.3) | 563–992 | 103 | 164.5 (4.6) | 153.5–176.8 | 103 | 27.4 (2.4) | 22.0–35.8 |
| Females | 46 | 780 (92.5) | 600–1005 | 46 | 162.1 (4.9) | 152.0–171.4 | 46 | 29.6 (2.8) | 22.8–34.9 | 70 | 719 (56.7) | 563–963 | 68 | 163.8 (4.5) | 153.5–173.4 | 68 | 26.8 (2.2) | 22.0–35.2 |
| Pregnant | 37 | 792 (91.6) | 600–1005 | 37 | 162.3 (5.0) | 152.0–171.4 | 37 | 30 (2.7) | 22.8–34.9 | N/A | N/A | N/A. | N/A. | N/A. | N/A. | N/A. | N/A. | N/A. |
| Not Pregnant | 9 | 732 (84.9) | 645–844 | 9 | 161.1 (4.4) | 155.1–168.4 | 9 | 28.2 (2.8) | 24.1–32.6 | 70 | 719 (56.7) | 563–963 | 68 | 164 (4.5) | 154.0–173.0 | 68 | 26.8 (2.2) | 22.0–35.2 |
| Males | 44 | 800 (105.8) | 585–1,008 | 44 | 164.6 (4.8) | 155.1–173.1 | 44 | 29.4 (3.0) | 22.6–36.1 | 35 | 787 (82.9) | 622–992 | 35 | 165.8 (4.6) | 158.1–176.8 | 35 | 28.6 (3.1) | 24.0–35.8 |
| Total Females | 86 | 710 (105.5) | 516–1,005 | 85 | 159.4 (5.4) | 147.0–171.4 | 85 | 27.9 (3.1) | 20.2–34.9 | 102 | 646 (134.7) | 291–963 | 99 | 160 (8.5) | 140.0–173.0 | 99 | 25.1 (3.7) | 13.7–35.1 |
| Total Males | 57 | 761 (119.9) | 559–1,008 | 57 | 162.8 (5.7) | 147.8–173.1 | 57 | 28.6 (3.2) | 22.6–36.1 | 56 | 660 (187.4) | 266–992 | 56 | 158.5 (11.1) | 127.3–176.8 | 56 | 25.7 (4.6) | 16.4–35.8 |
| TOTAL | 143 | 730 (113.8) | 516–1,008 | 142 | 160.7 (5.8) | 147.0–173.1 | 142 | 28.2 (3.1) | 20.2–36.1 | 158 | 651 (155.0) | 266–992 | 155 | 159.2 (9.5) | 127.3–176.8 | 155 | 25.3 (4.0) | 13.8–35.8 |

Seasonal demography statistics (number (n), mean, standard deviation (SD), range, winter and summer) of 301 Grey-headed flying foxes captured over six surveys between September 2015 and February 2018 from the Adelaide Camp, Botanic Park, Adelaide. BW = Body weight, FAL = forearm length, BCI = body condition index = $1,000^*(BW/FAL)^2$

## Hurdle modelling of seropositivity and antibody levels

Using lower threshold levels, the probability of Hendra virus seropositivity was positively and significantly associated with body condition index (Odds ratio = 3.67, p = 0.002). Cedar virus seropositivity was not associated with any of the investigated factors. 95.7% of all bats were Tioman virus seropositive and the hurdle model could not converge because of saturation (Table 4 and Fig 3). Using the antibody level model, Hendra virus antibody MFI levels were significantly higher in pregnant seropositive females and at the second survey in February 2016 (when 57.4% of individuals were seropositive). However, Hendra virus antibody MFI levels were significantly lower at the fifth survey in September 2017 when 37.0% of individuals were seropositive. Tioman virus MFI antibody levels were significantly higher in adults than sub-adults and at the fifth catching session in September 2017 (Table 4 and Fig 3). There was no evidence of co-seropositivity among the four viruses.

## Investigation of freedom from Australian bat lyssavirus seroprevalence

None of the tested bats yielded a MFI high enough to imply seroconversion. Accounting for the number of bats captured at each sampling session, there was enough evidence to demonstrate, with 95% confidence, that the Australian bat lyssavirus sero-prevalence is less that 2%, assuming that the probability of the camp to be exposed (or an immigrating bat being exposed) between samplings was ≤ 5% (Table 5). There was not enough evidence to demonstrate with confidence a seroprevalence ≤ 1% regardless of the probability of exposure.

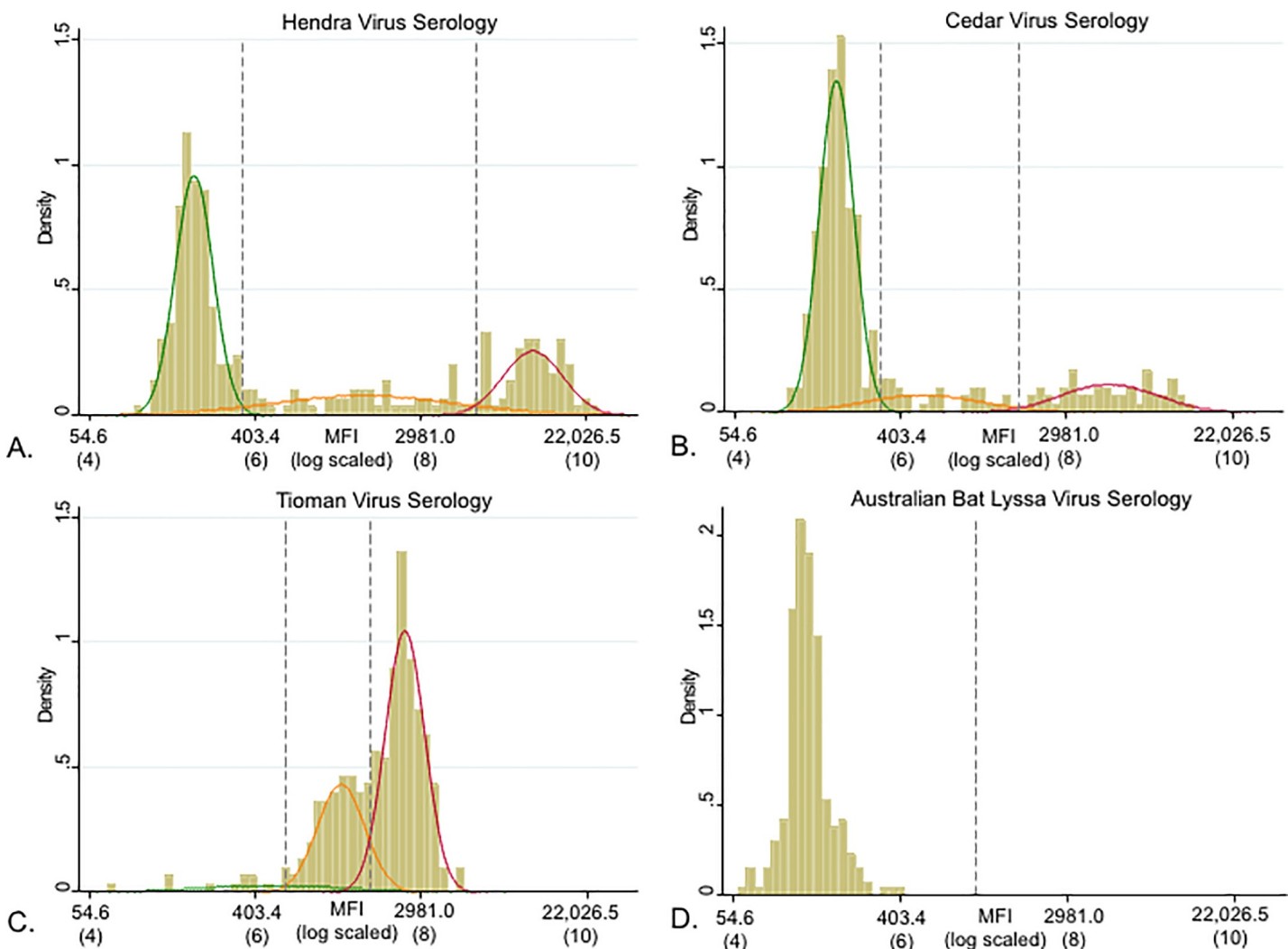

**Fig 2. Density histogram and overlaid mixture of modelled distributions.** Density histogram and overlaid mixture of modelled distributions for MFI and the natural log MFI of A) Hendra virus, B) Cedar virus, C) Tioman virus and D) Australian bat lyssavirus serological readings. Thresholds (dashed vertical lines) correspond to the intersection between a pair of predicted distributions. Upper and lower thresholds for the Hendra virus serology were determined as the natural antilogorithm of 5.85 and 8.67, respectively (MFI 347 and 5,825, respectively). Upper and lower thresholds for the Cedar virus serology was determined as the natural antilogorithm of 5.76 and 7.44 respectively (MFI 317 and 1,702 respectively). Upper and lower thresholds for the Tioman virus serology was determined as the natural antilogorithm of 6.37 and 7.38 respectively (MFI 584 and 1,603 respectively). Upper and lower thresholds could not be calculated for Australian bat lyssavirus as it was determined that all animals belong to the same exposure sub-population.

**Table 3. Lower threshold MFI scores with corresponding % seroprevalence.**

| Virus | Lower MFI threshold (log MFI) | % seroprevalence (Binomial exact 95% CI) | MFI median values (range) for seropositive animals |
|---|---|---|---|
| Hendra virus | 347 (5.85) | 43.2% (37.5%-49%) | 6,813 (353–23,922) |
| Cedar virus | 317 (5.76) | 26.6% (21.7%-31.9%) | 3,074 (326–13,759) |
| Tioman virus | 584 (6.37) | 95.7% (92.7%-97.7%) | 2,121 (629–4,972) |
| Australian bat lyssavirus | Na | 0% (0.0%- 1.22%) | Na |

Lower thresholds including median fluorescence intensity (MFI) and log MFI and seroprevalence with confidence intervals (CI) for Hendra virus, Cedar virus, Tioman virus and Australian bat lyssavirus for Grey-headed flying foxes sampled in Adelaide, South Australia between September 2015 and February 2018 (n = 301). na = not applicable.

**Table 4. Statistics for hurdle and antibody level models.**

| MODEL | Hendra virus | | | | | | Cedar virus | | | | | | Tioman virus | | | | |
|---|---|---|---|---|---|---|---|---|---|---|---|---|---|---|---|---|---|
| HURDLE | Estimate | OR (95% CI) | SE | Z value | P value | | Estimate | OR (95% CI) | SE | Z value | P value | | Estimate | OR (95% CI) | SE | Z value | P value |
| Intercept | -0.311 | - | 0.204 | -1.526 | 0.127 | | -0.996 | - | 0.221 | -4.5 | <0.001 | | . | . | . | . | . |
| Adult male | -0.036 | -1.04 | 0.308 | -0.118 | 0.906 | | -0.298 | -1.34 | 0.352 | -0.847 | 0.397 | | . | . | . | . | . |
| Female Not Preg | 0.04 | 1.04 | 0.313 | 0.127 | 0.899 | | 0.407 | 1.5 | 0.33 | 1.235 | 0.217 | | . | . | . | . | . |
| Female Preg | 0.057 | 1.05 | 0.379 | 0.151 | 0.88 | | -0.587 | -1.8 | 0.471 | -1.246 | 0.213 | | . | . | . | . | . |
| BCI centred | **1.3** | **3.67** | **0.416** | **3.126** | **0.002** | | -0.27 | 1.31 | 0.451 | -0.598 | 0.55 | | . | . | . | . | . |

| ANTIBODY LEVEL | Estimate | SE | Z value | P value | Estimate | SE | Z value | P value | Estimate | SE | Z value | P value |
|---|---|---|---|---|---|---|---|---|---|---|---|---|
| Intercept | 8.717 | 0.218 | 40.027 | <0.001 | 7.616 | 0.347 | 21.96 | <0.001 | 7.543 | 0.07 | 108.318 | <0.001 |
| Adult male | 0.067 | 0.228 | 0.295 | 0.768 | 0.365 | 0.308 | 1.185 | 0.236 | **0.227** | **0.062** | **3.667** | **<0.001** |
| Female Not Preg | -0.15 | 0.205 | -0.732 | 0.464 | 0.133 | 0.316 | 0.42 | 0.675 | **0.161** | **0.068** | **2.374** | **0.018** |
| Female Preg | **0.865** | **0.281** | **3.073** | **0.002** | 0.797 | 0.446 | 1.787 | 0.074 | **0.394** | **0.084** | **4.685** | **0** |
| BCI centred | 0.346 | 0.315 | 1.099 | 0.272 | -0.113 | 0.461 | -0.245 | 0.806 | 0.097 | 0.091 | 1.064 | 0.287 |
| 16-Feb | **0.715** | **0.258** | **2.766** | **0.006** | 0.436 | 0.469 | 0.93 | 0.353 | -0.107 | 0.087 | -1.231 | 0.218 |
| 16-Aug | -0.218 | 0.295 | -0.739 | 0.46 | -0.045 | 0.489 | -0.092 | 0.927 | -0.066 | 0.086 | -0.761 | 0.447 |
| 17-Feb | -0.209 | 0.291 | -0.719 | 0.472 | 0.78 | 0.43 | 1.813 | 0.07 | -0.122 | 0.091 | -1.341 | 0.18 |
| 17-Aug | **-0.583** | **0.271** | **-2.152** | **0.031** | 0.093 | 0.477 | 0.194 | 0.846 | **-0.192** | **0.088** | **-2.191** | **0.028** |
| 18-Feb | 0.374 | 0.336 | 1.113 | 0.266 | 0.645 | 0.531 | 1.213 | 0.225 | 0.087 | 0.096 | 0.907 | 0.365 |

Odds ratios (OR), estimates, standard errors (SE), Z values and P values for negative binomial hurdle and antibody level models for Hendra virus, Cedar virus and Tioman virus serology for Grey-headed flying foxes sampled between August 2015 and February 2018. Reference values relate to sub-adults (both male and female) for the hurdle and antibody level model and for the first catching session (August 2015) for the antibody level model. Preg = pregnant. BCI centred = body condition index centred around the mean values. CI = confidence interval.

### Recapture seroprevalence analysis

Over the six sessions, four bats were recaught; three males and one female (Table 6). Between survey one and two, September 2015 and February 2016, respectively, the Hendra virus MFI antibody level for one male almost doubled from MFI 9428 to 16929, suggesting exposure occurred prior to September 2015 and continued until February 2016 or reinfection or recrudescence of Hendra virus occurred during this same period. This male's weight also increased as it was classified as a sub-adult in September 2015 and an adult in February 2016. Another male seroconverted for Cedar virus between Sept 2015 and Feb 2016, suggesting exposure occurred during that period. All four animals were seropositive for Tioman virus at both sampling periods while two males did not seroconvert for Hendra virus and thus remained seronegative between the two six month time periods. Furthermore, two males and one female did not show evidence of exposure to Cedar virus between sampling periods.

## Discussion

Our study showed strong evidence of exposure of Adelaide Grey-headed flying fox camp to Hendra virus, Cedar virus and Tioman virus and no evidence of exposure to Australian bat lyssavirus. The semiquantitative results provided by Luminex binding assays also identified individual-level correlates of seropositivity and antibody levels. Hendra virus seroprevalence in this study (43.2%, 95%CI: 37.5%-49%) is similar to that reported previously (44.5%) [31] using a Luminex binding assay and compares with an overall seroprevalence of 23.6% using a serum

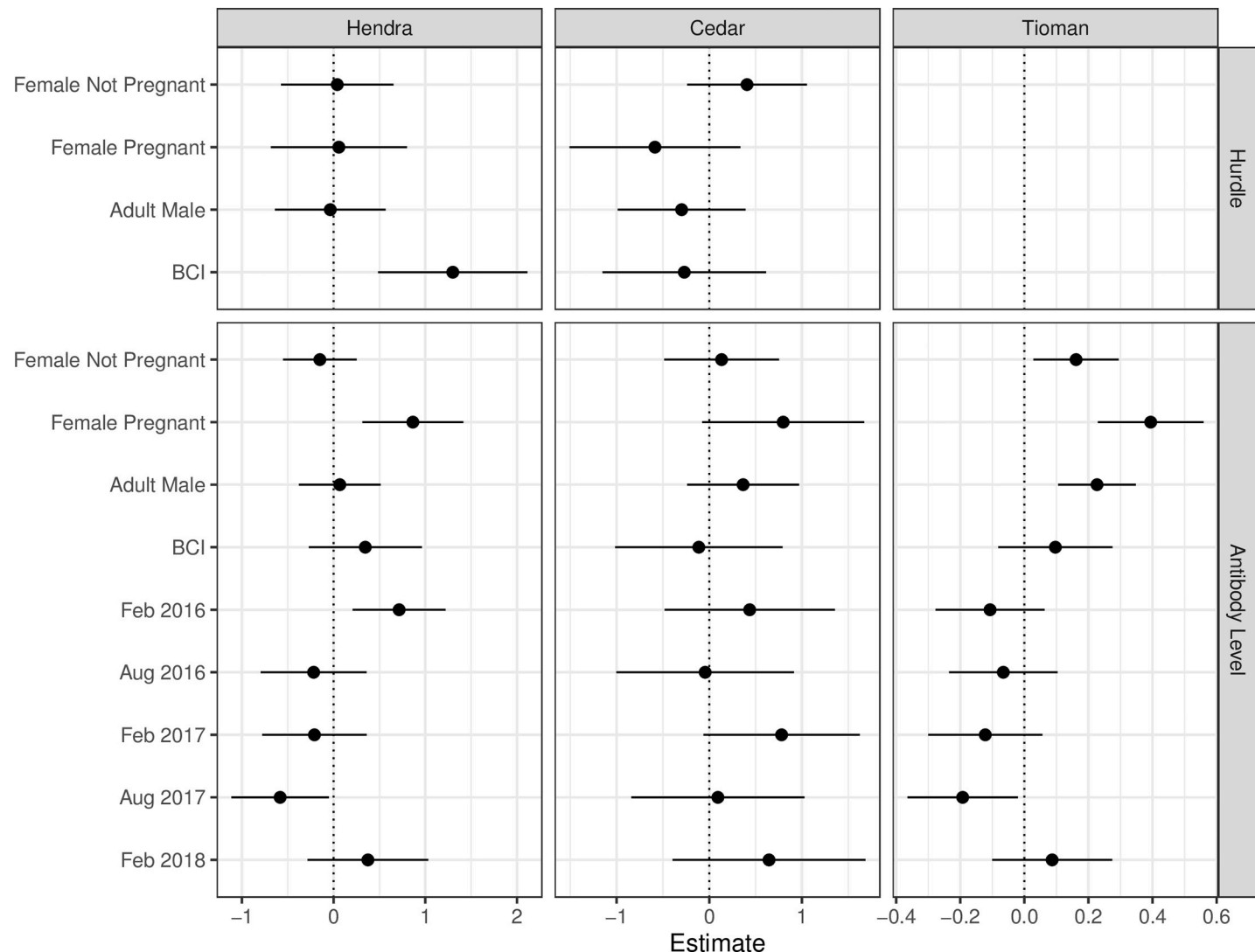

**Fig 3. Effect plots for hurdle and antibody level models.** Effect plots for the estimates and P values for negative binomial hurdle and antibody level models for Hendra virus, Cedar virus and Tioman virus serology for Grey-headed flying foxes sampled between August 2015 and February 2018.

neutralisation test in Little red flying foxes (*P.scapulatus*) [23] and an overall seroprevalence of 56% using a serum neutralisation test in Spectacled flying foxes [21]. Cedar virus seroprevalence was half than previously reported; 26.6% (95%: 21.7%-31.9%) versus 51.1% in Grey-

**Table 5. Probablity of freedom from Australian bat lyssavirus exposure.**

| Design prevalence (P*) | Probability of exposure | | | | |
|---|---|---|---|---|---|
| | **0.5%** | **1%** | **2%** | **5%** | **10.0%** |
| 1% | 94.6% | 93.8% | 92.2% | 87.2% | 78.4% |
| 2% | **99.5%** | **99.2%** | **98.6%** | **96.7%** | 93.3% |
| 5% | **100.0%** | **99.9%** | **99.8%** | **99.6%** | 99.1% |
| 10% | **100.0%** | **100.0%** | **100.0%** | **100.0%** | **99.9%** |

Summary of final probability of freedom from Australian bat lyssavirus exposure after 6 surveys of a total 301 bats (all seronegative) from the Adelaide Grey-headed flying fox camp. Bold types represent values of probability of freedom where a minimum threshold of 95% was reached.

**Table 6. Seroprevalence changes in recaptured Grey-headed flying foxes.**

| Bat ID | Date of capture | Sex | Wt /g | BCI | Hendra virus MFI | Hendra virus seropositive | Cedar virus MFI | Cedar virus seropositive | Tioman virus MFI | Tioman virus seropositive | Australian bat lyssavirus MFI | Australian bat lyssavirus seropositive |
|--------|-----------------|-----|-------|-----|------------------|---------------------------|-----------------|--------------------------|------------------|---------------------------|-------------------------------|----------------------------------------|
| 6 & 72 | 31 Aug 2015 | F | 844 | 29.8 | 772 | - | 139 | - | 2797 | + | 136 | - |
|        | 26 Feb 2016 |   | 763 | 26.9 | 745 | - | 153 | - | 2355 | + | 91 | - |
| 23 & 70 | 2 Sept 2015 | M | 820 | 30.1 | 172 | - | 181 | - | 2592 | + | 130 | - |
|        | 26 Feb 2016 |   | 744 | 26.6 | 128 | - | 3073 | + | 1248 | + | 115 | - |
| 46 & 81 | 3 Sept 2015 | M | 666 | 25.4 | 9428 | + | 175 | - | 2820 | + | 92 | - |
|        | 26 Feb 2016 |   | 773 | 28.0 | 16929 | + | 191 | - | 2979 | + | 141 | - |
| 145 & 252 | 10 Aug 2016 | M | 818 | 30.7 | 8710 | + | 257 | - | 2217 | + | 204 | - |
|        | 13 Aug 2017 |   | 830 | 30.8 | 6335 | + | 210 | - | 3111 | + | 207 | - |

The sex, weight (Wt), body condition index (BCI) and study identification number (Bat ID) of four Grey-headed flying foxes recaptured between August 2015 and August 2017 at the Adelaide Camp, Adelaide, South Australia and their median fluorescence intensity (MFI) serostatus for Hendra virus, Cedar virus, Tioman virus (using lower thresholds) and Australian bat lyssavirus. M = Male; F = Female; Seropositive = +; Seronegative = —.

headed flying foxes [31]. There were some differences in the exposure rates within the camp and across the study sampling times.

Hendra seropositivity was also positively associated with BCI. This contrasts with a previous study [23] which found increased seropositivity in nutritionally-stressed Little-red flying foxes but concurs with a study [10] in Black flying foxes and Grey-headed flying foxes. In our study, the body condition index of bats was significantly higher in winter than summer (Table 2). Food quantity and quality for Grey-headed flying foxes are usually inferior in winter elsewhere in their normal range [50]. Winter immigration of approximately 5–10,000 extra Grey-headed flying foxes into the Adelaide camp (Van Weenan pers comm) in 2018 and 2019 suggests that Adelaide is an attractive feeding ground during winter. Other studies indicate acute food shortages may be associated with El Nino/La Nina climate cycles [51] leading to nutritionally stressed animals and this may be the driver for the seasonal patterns of Hendra virus seroprevalence [23, 52]. Late gestation was positively associated with higher Hendra virus MFI antibody levels in comparison to non-pregnant females and males. SImilar evidence is seen in serological surveys of Spectacled flying foxes [21] and Little-red flying foxes [23] which showed increased detection of Hendra virus antibodies associated with late-stage gestation or early lactation but is in contrast to recent research in Grey-headed flying foxes [10] where there is no association.

Hendra virus and Tioman virus seropositivity varied across surveys (Fig 3) with Hendra virus seroprevalence significantly increasing between August 2015 and February 2016. This pattern could be explained by: i) "exposure and spread in a sedentary camp" where a Hendra virus exposure event that occurred before August 2015 (seroprevalence = 49%) and resulted in an increase in seroprevalence of captured animals in February 2016 (seroprevalence = 57%) without any further exposure occurring in this period (i.e. within camp spread) and negligible emigration/immigration; or ii) "exposure and re-exposure in a sedentary camp" where additional Hendra virus exposure occurred between the two sampling periods which led to an

higher seroprevalence at the second sampling period and negligible emigration/immigration; or iii) "periodic emigration" of non-exposed animals and/or "periodic immigration" of previously exposed flying foxes occurred during this period. Previous studies have suggested that Hendra virus is maintained in flying fox populations through episodic infection in a metapopulation structure, and do not persist endemically within a single population [23]. Most hypotheses emphasize horizontal transmission within colonies via urine and other secretions, especially during pregnancy and mating [23], or via migration, with the magnitude of migration affected by the spatial connectivity among colonies, resulting in episodic infection [53].

The recaptures of four individuals over the sampling period provided some information on the immunity dynamics of these viruses within this specis. Two of these animals were not exposed to Hendra virus and thus remained seronegative between the two captures (six month time period for both). However, one animal's Hendra virus seropositive MFI antibody level nearly doubled over a six month time period; between September 2015 and February 2016 which could mean: i) it was recently exposed just before the first sampling and the antibody level continued to rise in response to the second sampling; ii) it was exposed some time before the first sampling and the antibody level peaked between the two captures and was waning at the second; or iii) it was exposed some time before the first sampling and was re-exposed between captures and mounted a further antibody response. Epstein et al, 2013 [30] suggests maternal antibodies to Hendra virus in Black flying foxes last between 7.5 and 8.5 months and acquired immunity to African henipaviruses may last up to 4 years in adult *Eidolon helvum* adults [54] but evidence on Grey-headed flying foxes immune response to viruses is sparse. The antibody level of another seropositive animal waned over one year between August 2016 and 2017, suggesting that the animal was less likely to have become further infected (August 2016 to August 2017). The fourth animal was seropositive and its antibody level waned over one year (August 2016 to August 2017), suggesting that this animal was unlikely re-exposed during this period.

None of the explored explanatory variable predicted Cedar virus serostatus which is consistent with previous reports [31]. Furthermore there was no evidence of association between the serostatuses of any pair of viruses. Adult bats showed significantly higher antibody levels against Tioman virus in comparision to subadults which may suggest there is a cumulative age-related antibody response to multiple exposures of the virus. Additionally, immunofluorescent antibody and immunoelectron microscopic data suggested that Tioman virus is antigenically related to Menangle virus [12] so it is possible that the high seropositivity to Tioman virus could result from the cross reactivity with Menangle virus exposure.

No evidence of Australian bat lyssavirus exposure was found over our study period despite a previous finding of a positive diagnosis in a Grey-headed flying fox in the Adelaide camp in September, 2012 [20]. Previous serological surveys have found a 3.0% Australian bat lyssavirus seroprevalence in flying foxes (95% CI: 1.5–5.8%)[55] using the rapid fluorescent focus inhibition test and 2.9% seroprevalence (95% CI: 1.8–4.5%) in six insectivorous species in Western Australia using a Luminex multiplex binding assay [33]. Rabies virus neutralising antibodies have been shown to wane in experimentally-infected bats within 6 months after an initial inoculation, but persisted for longer (6–12 months) after a second inoculation of surviving bats [56]. Our results suggest that either (i) Australian bat lyssavirus has not been circulating in the camp over this time period; (ii) seropositivity is very short lived; or (iii) infected flying foxes died suddenly and were thus not sampled at surveys. However, bats are thought to be the ancestral reservoir of lyssaviruses [57] and are the only taxa in which antibodies are detected with sufficient frequency to support serosurveillance [33] which could indicate that the virus is unlikely to be circulating in the Adelaide camp.

As with all flying fox camps, the population dynamics can often be very fluid with regular patterns of immigration, emigration and range expansions. Some studies show flying foxes can

travel hundreds of kilometres [15, 55, 58], moving regularly between different camps over their distribution range. Furthermore, there is evidence that all four species of mainland Australian pteropids can co-occur in the same camp [59]. Range expansions and contractions have been noted in both Black flying foxes and Grey-headed flying foxes [18, 60–61]. The range of Black flying foxes has increased southwards greater than 1000km during the twentieth century [62] and this has been proposed as a possible contributing factor to a contraction of Grey-headed flying foxes distribution range. In its southernmost distribution, Grey-headed flying foxes now live in the urban environments of Melbourne [61] and Adelaide. While these areas are not thought to be part of the 'climatic niche' of the species during winter, increased temperature due to the 'urban heat island effect' and climate change may have created an environment that is now tolerable [63]. Therefore its conceivable that through this overlap of flying fox species, transmission and infection may occur anywhere along the distribution range continuum at any time.

Microsphere assays provide a sensitive method to detect henipavirus antibody binding in fruit bat plasma and serum [15, 32, 36]. The output of these assays, median fluorescence intensity (MFI), are continuous data and present a challenge in determining meaningful threshold values that categorise bats as seropositive or seronegative [36]. A MFI > 1,000 for Australian bat lyssavirus has been considered positive and < 250 negative [33]. Our use of mixture models to determine threshold values reflects that of Burroughs et al 2016 [31] in that we accept that a single threshold is not possible for the serological profile obtained for the Adelaide bats. We looked for 'natural' groupings of binding activity and used two threshold values to divide these groups into negative, intermediate and positive categories. We recognise that binding in the intermediate category may represent an important intermediate stage in antiviral protection, the shift from a seronegative to a seropositive state or vice versa, or may represent a susceptible state. Even using the more specific threshold (MFI 5825, 1702, 1603 for Hendra virus, Cedar virus and Tioman virus, respectively), 25.2% of bats caught from the Adelaide camp showed evidence of prior infection with Hendra virus, 16.6% with Cedar virus and 63.8% with Tioman virus which all suggest common exposures at both the individual and camp level.

We acknowledge certain limitations to our study. The most effective technique to capture bats in the Adelaide camp requires nets to be placed from suitable trees under the camp as they return from foraging. The entire foot print of bat roost trees could not be sampled using a formal random sampling approach because of the topography and may therefore consistute a potential sampling bias. Utilising the same capture sites across the whole study period attempted to standardise any potential sampling bias and protect the comparibility of samples.

## Conclusion

In contrast to other studies, good body condition rather than nutritional stress was an indicator of increased Hendra virus seroconversion. Substantiating other studies, Hendra virus antibody levels were higher in pregnant females. Unexpectedly, there was no evidence of Australian bat lyssavirus seroconversion. This study highlighted the successful use of a multiplexed Luminex binding assay for serological surveys in flying foxes but also the need to expand the research to include more sampling periods over an annual cycle and to compare with viral presence and diversity.

## Supporting information

**S1 Table. Catching information, demographic data, morphometrics and MFI values for Hendra virus, Cedar virus, Tioman virus and Australian bat lyssa virus for each sampled bat.**
(XLSX)

## Acknowledgments

We thank all the volunteers who supported the capture and processing of flying foxes, especially staff from the SA Department of Environment and Water, academic colleagues, students and volunteers. A special thanks goes to Clive Boardman, Dr Michelle Power, Dr Cecilia Sanchez, and Dr Kathy Burbidge for long term support and expertise and the staff of Zoos SA for their ongoing support including Drs David and Jenny McLelland, Dr Jerome Kalvas and Dianne Hakof and Rebecca Probert. We acknowledge Dr Christopher Broder and Dr Lianying Yan, Uniformed Services University, USA for the production of the Hendra virus and Cedar virus soluble glycoproteins and Dr Kestas Sasnauskas, Institute of Biotechnology, Lithuania for the production of the Tioman virus nucleocapsid protein used in this study. Wildlife health data were provided by the Wildlife Health Australia Coordinator in South Australia and generated by the Department of Primary Industries and Regions, South Australia.

## Author Contributions

**Conceptualization:** Gary Crameri, Charles G. B. Caraguel, Thomas A. A. Prowse.

**Data curation:** Charles G. B. Caraguel.

**Formal analysis:** Charles G. B. Caraguel, Thomas A. A. Prowse.

**Investigation:** Michelle L. Baker, Victoria Boyd, Gary Crameri, Terry Reardon, Ian G. Smith.

**Methodology:** Victoria Boyd, Gary Crameri, Grantley R. Peck, Terry Reardon, Ian G. Smith, Charles G. B. Caraguel, Thomas A. A. Prowse.

**Resources:** Michelle L. Baker, Victoria Boyd, Terry Reardon, Ian G. Smith.

**Supervision:** Charles G. B. Caraguel, Thomas A. A. Prowse.

**Validation:** Michelle L. Baker, Victoria Boyd, Grantley R. Peck, Charles G. B. Caraguel, Thomas A. A. Prowse.

**Writing – original draft:** Wayne S. J. Boardman.

**Writing – review & editing:** Wayne S. J. Boardman, Michelle L. Baker, Victoria Boyd, Grantley R. Peck, Terry Reardon, Ian G. Smith, Charles G. B. Caraguel, Thomas A. A. Prowse.

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
