## [Decision Letter · Decision Letter 0]

6 Dec 2019

PONE-D-19-30981

Seroprevalence of three paramyxoviruses; Hendra virus, Tioman virus, Cedar virus and a rhabdovirus, Australian bat lyssavirus, in a range expanding fruit bat, the grey-headed flying fox <(Pteropus poliocephalus)>.

PLOS ONE

Dear Dr Boardman,

Thank you for submitting your manuscript to PLOS ONE. After careful consideration, we feel that it has merit but does not fully meet PLOS ONE’s publication criteria as it currently stands. Therefore, we invite you to submit a revised version of the manuscript that addresses the points raised during the review process. by two reviewers. 

In addition to these comments please make all meta data available  either as part of the manuscript or supporting information or deposit this into a public repository and provide the link to this information. 

We would appreciate receiving your revised manuscript by Jan 20 2020 11:59PM. To enhance the reproducibility of your results, we recommend that if applicable you deposit your laboratory protocols in protocols.io, where a protocol can be assigned its own identifier (DOI) such that it can be cited independently in the future. For instructions see: http://journals.plos.org/plosone/s/submission-guidelines#loc-laboratory-protocols

We look forward to receiving your revised manuscript.

Kind regards,

Wanda Markotter

Academic Editor

PLOS ONE

Journal Requirements:

2. We note that you have reported significance probabilities of 0 in places. Since p=0 is not strictly possible, please correct this to a more appropriate limit, eg 'p<0.0001'.

3. We note that [Figure(s) 1] in your submission contain [map/satellite] images which may be copyrighted. All PLOS content is published under the Creative Commons Attribution License (CC BY 4.0), which means that the manuscript, images, and Supporting Information files will be freely available online, and any third party is permitted to access, download, copy, distribute, and use these materials in any way, even commercially, with proper attribution. For these reasons, we cannot publish previously copyrighted maps or satellite images created using proprietary data, such as Google software (Google Maps, Street View, and Earth). For more information, see our copyright guidelines: http://journals.plos.org/plosone/s/licenses-and-copyright.

1.    You may seek permission from the original copyright holder of Figure(s) [1] to publish the content specifically under the CC BY 4.0 license. 

4. In your Methods section, please provide additional location information, including geographic coordinates for the data set if available.

Reviewers' comments:

Reviewer's Responses to Questions

**Comments to the Author**

1. Is the manuscript technically sound, and do the data support the conclusions?

Reviewer #1: Yes

Reviewer #2: Yes

2. Has the statistical analysis been performed appropriately and rigorously? 

Reviewer #1: Yes

Reviewer #2: Yes

3. Have the authors made all data underlying the findings in their manuscript fully available?

Reviewer #1: No

Reviewer #2: No

4. Is the manuscript presented in an intelligible fashion and written in standard English?

Reviewer #1: Yes

Reviewer #2: Yes

5. Review Comments to the Author

Reviewer #1: Review of ‘Seroprevalence of three paramyxoviruses; Hendra virus, Tioman virus, Cedar virus, and a rhabdovirus, Australian bat lyssavirus, in a range expanding fruit bat, the Grey-headed flying fox (Pteropus poliocephalus)

The authors present data on the seroprevalence of bat borne paramyxoviruses through longitudinal sampling using a microsphere multiplex serological assay using surface proteins as targets. They analyze the serological data using finite mixture models on log-transformed MFI values to determine distributions and assign categories of seronegative, seropositive or intermediates and then use hurdle modelling to correlate seropositivity and titre (MFI value). As more serological datasets are generated from species with no true positive or negative controls, it’s important that there are advanced analysis of these data. Longitudinal sampling is needed to understand temporal patterns of virus periodicity and, by proxy, relative risk. Setting up and conducting longitudinal surveillance projects requires significant foresight and a significant allocation of time, expenses and personnel. The manuscript is well written and presents compelling data on the seroprevalence of multiple bat paramyxoviruses. There are revisions required before this manuscript can be accepted.

Major revisions

• I believe that the introduction and discussion can be streamlined. The introduction is 8 paragraphs and the discussion is 9 paragraphs. There is much salient data in these sections, but they should be cut down for the benefit of the reader.

• The methods section for the serology, which plays an integral role in the study results, is not detailed enough. The authors should provide a significantly more detailed section including; the proteins (target and how they are produced, if samples were run in duplicate/triplicate, what machine they were run on. I understand that the authors are referencing a previous study, but I imagine details in the bat handling and serum collection could be similarly truncated with a reference to similar approaches. Additionally, reference 31 in this section is for nucleic acid detection and not antibody detection and needs to be removed.

• Have the authors looked at the significance of using weight only as opposed to using body condition index when studying seroprevalence. A meta-analysis done by McGuire et al 2018 (https://doi.org/10.1093/jmammal/gyy103) found that there was no support that, “bats with longer forearms weigh more than bats with shorter forearms”. They note that the relationship between body mass and forearm length was very weak. Is this the case in Pteropus poliocephalus and is BCI an appropriate metric to study?

• The only reference to raw MFI values was in the recapture data. There was no supplementary table with the MFI values. The authors should include this data and it would help if they added a scatterplot or box plot with these values for each virus glycoprotein with the lower and upper thresholds noted. Or they can add a secondary x-axis on figure 2 to know what the MFI values are. This is important because these multiplex serological assays are being implemented on pteropodid bats across South and Southeast Asia and it’s important for comparative studies to know what raw MFI values are for seropositives and seronegatives.

• The authors should be cautious when they use the word antibody titres in the text and in the methods and results section headers. They state that antibody titre is expressed as MFI, but titre has a specific meaning. Have the authors run an endpoint dilution to then correlate that endpoint with MFI detection values? If they have not, then they should not use the word titre.

Minor revisions

Abstract

• Line 37: What does ‘camp antibody’ titres mean?

Introduction

• Lines 52&53: Check duplicate references

• Line 72: Is there a reference for the range expansion of the species?

• Line 76: What does ‘incidentally detected’ mean?

• Line 82: Please use another word for elusive

• Line 88-89: The authors should mention the conformational state of the proteins in these tests as native state antigens will represent what the immune systems sees compared to linear peptides.

• Line 97: The authors use both recrudescence and reactivation in this manuscript. Please use one or the other for consistency.

Materials and Methods

• Line 144: What is the company that makes the mist nets?

• How is a subadult defined?

• Is estimated age a subjective categorical variable? Please explain.

• How was the sera treated before the test? Was it heat inactivated? Was is gamma inactivated?

• Line 169: Was the concentration of antibodies measured? It is better to state that antibodies were detected against the different viruses in the panel

• Line 172: Please remove reference 31 as it does not refer to antibodies

• Line 183 section: Were any priors used for Hendra and Cedar virus when running the models? There were priors used and noted for Australian bat lyssavirus, but none are mentioned for these two henipaviruses, even though there are previous studies studying seroprevalence. Noted in line 339-343

• Line 196: Can you please explain how threshold values were ‘visually determined’?

Results

• In the demographic section, some description of the collections would be helpful (adults compared to sub adults and males compared to females). I understand this is noted in the table, but some more detail in the text would be helpful.

• The rest of the results reads well.

Discussion

• Are there specific periods of birthing in this species? There is a lot of natural history information, but I did not note this important fact.

• Line 376: The authors note that their findings support the SIRS and SILI models. Can they expand on this statement in this paragraph.

• Line 401: Why wasn’t there an association between co-serostatus?

• Line 413: The authors can reference Peel et al’s work on Africa henipavirus immunity duration (DOI: 10.1038/s41598-018-22236-6) and Schuh et al 2019 (Comparative analysis of serologic cross-reactivity using convalescent sera from filovirus-experimentally infected fruit bats) and this would support their discussion on cross-reactivity and decay in filoviruses.

• Line 438-439: There are references to specific MFI values, but there are for both henipaviruses and lyssaviruses. This sentence needs to be more detailed as to which MFI values correspond to what specific viruses.

• Line 446: The author’s reference the specific thresholds for MFI values and having an additional figure with the non-transformed MFI values would help.

• Line 454-456: was there a reason why it wasn’t possible to collect urine to establish virus prevalence?

• Line 466-467: The authors state that there is no significant co-occurance of the studied viruses, but they did not test for viruses. They only studied seroprevalence.

Table 2:

• It would be beneficial to have the range of MFI values and the median values for seropositives in this table.

Figures

• Figure 1 looks to be from google maps. That satellite image from a copyrighted source may not be publishable under the PLOS Open Access CC-BY License

Reviewer #2: This manuscript describes a three year serological study of grey-headed flying-foxes at a single roost site in South Australia. Tests on sera collected were conducted for evidence of exposure to four viral pathogens and analyses performed to identify risk factors for exposure and patterns of infection.

The text is generally very well written and the key points are clearly communicated, however I have some concerns with the interpretation of the results and their significance.

Major comments:

1 – There are various terms used to describe the study site and the group of animals sampled - “colony” “flying fox camp” “Adelaide grey-headed flying fox population”. Clarification of the terms used is necessary to interpret the results and understand their significance.

2 - From an epidemiological perspective – what is the study population? i.e. which group of flying foxes constituted the sampling frame from which the sample were drawn?

The text includes “immigration of approximately 10,000 extra grey-headed flying foxes into the Adelaide camp”. Where did these bats come from? The reader needs to have a clear understanding of the group of flying foxes that could have been selected for sampling in this study, e.g. to understand the potential bias in only sampling from one location.

3 – what is the target population? i.e. to which group of flying-foxes are the results of this study to be applied? This is relevant to clarify when considering which previous studies are appropriate to include for comparison of results in the Discussion. And to consider the application and relevance of these results to other flying-fox populations/roosts/species etc.

The term “Adelaide colony” is used frequently in the text but it seems evident that the group of bats present at the roost site in Adelaide does not constitute a “colony” in the biological sense of the word. “Roost” or “camp” site would seem more appropriate given the evidence of long-distance travel and high level of genetic mixing of this species. I understand the two questions above may be difficult, or not possible, to answer with high confidence. But if this is the case, then this should be explicitly explained and discussed. Perhaps two or more hypotheses could be presented for potential study and target populations, and then the results and conclusions considered in light of different potential population definitions.

Regarding the demonstration of freedom of infection from ABLV:

4 - The stated assumption of perfect accuracy of the Luminex assay seems unreasonable without strong supporting evidence. Methods are readily available for analysis of results with assumptions of imperfect sensitivity and specificity of a diagnostic test and this would seem more reasonable.

5 - There is a statement in the Discussion that it is possible the “antigen used in the Luminex assay no longer contains a relevant epitope”. Further work to clarify this would seem necessary to interpret the results. How does the antigen in the assay compare to those found on the corresponding virus detected in Adelaide in 2012?

6 – The freedom from infection calculations require clarification or greater explanation in the context of clarification of the definition of the “Adelaide colony” i.e. the target and study populations. Which group of bats is being considered free from infection? Are the “10,000 extra bats” included in this?

7 – Given the statement “Freedom from Australian bat lyssavirus was deemed achieved if the estimated probability of freedom was ≥ 95%.” What does ≥ 95% mean in this context? How can this be useful?

8 – given the comments 4-7 above, I’m not convinced that demonstration of freedom from infection is very meaningful or useful in this context. Please provide explanation of the purpose in using the “freedom of infection” approach.

Minor comments:

9 – the discussion on infection dynamics and previous studies is useful but could be greatly improved through clarification of the target and study populations in this study. Currently it seems rather speculative.

10 – Re the statement “Uniquely, four recaptures of individuals over the sampling period provided some information on the infection dynamics of these viruses”. In what context is this unique?

Also, data from the recapture of such a small number of individuals seems more like interesting data to perhaps guide future studies rather than true information on infection dynamics. Particularly in light of the limitation of the tests used and uncertainty of the viruses to which the bats have been exposed.

11 – Re the statement “there is no evidence of how long titres might last in adults”. I think some of the previous studies referred to in the Introduction and Discussion have some evidence on this. At least with regards to potential models of infection dynamics and duration of immunity.

12 – it would be helpful to provide confidence intervals for the seroprevalences measured for ABLV in previous studies to aid interpretation of results from this study.

13 – re the statement “overlap of flying fox species coupled with their range expansions, transmission and infection may occur anywhere along the distribution range”. I do not see that range expansions are required for transmission and infection to occur throughout the range of Australian flying-fox species.

14 – re the statement “its possible that the high seropositivity to Tioman virus could reflect exposure due to either or both viruses at different times.” Given the evidence of Menangle virus and not Tioman virus presence in Australia, why test for Tioman antibodies and not Menangle antibodies? Or at least conduct a study to estimate the degree of cross reactivity of antibodies in this species.

15 – the statements “all four species of mainland Australian pteropids” and “one of five species of flying foxes found in Australia,” appear potentially contradictory, please clarify.

16 – lines 325-326 I think “did not become infected” should be “did not show evidence of exposure”

17 – regarding the statements on reasons for not sampling during October to December and April to May. How was the potential detrimental effect of sampling during these times determined? How is this reconciled with the stated conclusion to “include more sampling periods over an annual cycle”?

18 – please check for minor typographical errors e.g. lines 84-85, line 405 “its” should be “it is”

6. PLOS authors have the option to publish the peer review history of their article (what does this mean?). If published, this will include your full peer review and any attached files.

Reviewer #1: No

Reviewer #2: No

---

## [Author Response · Author response to Decision Letter 0]

12 Mar 2020

Please see response to reviewers table and cover letter

---

## [Decision Letter · Decision Letter 1]

14 Apr 2020

Seroprevalence of three paramyxoviruses; Hendra virus, Tioman virus, Cedar virus and a rhabdovirus, Australian bat lyssavirus, in a range expanding fruit bat, the grey-headed flying fox <(Pteropus poliocephalus)>.

PONE-D-19-30981R1

Dear Dr. Boardman,

We are pleased to inform you that your manuscript has been judged scientifically suitable for publication and will be formally accepted for publication once it complies with all outstanding technical requirements.

With kind regards,

Wanda Markotter

Academic Editor

PLOS ONE

Additional Editor Comments (optional):

Reviewers' comments:

Reviewer's Responses to Questions

**Comments to the Author**

1. If the authors have adequately addressed your comments raised in a previous round of review and you feel that this manuscript is now acceptable for publication, you may indicate that here to bypass the “Comments to the Author” section, enter your conflict of interest statement in the “Confidential to Editor” section, and submit your "Accept" recommendation.

Reviewer #1: All comments have been addressed

2. Is the manuscript technically sound, and do the data support the conclusions?

Reviewer #1: Yes

3. Has the statistical analysis been performed appropriately and rigorously? 

Reviewer #1: Yes

4. Have the authors made all data underlying the findings in their manuscript fully available?

Reviewer #1: Yes

5. Is the manuscript presented in an intelligible fashion and written in standard English?

Reviewer #1: Yes

6. Review Comments to the Author

Reviewer #1: The authors have addressed all the comments in this revised version. We thank them for providing a comprehensive set of edits that have improved the manuscript.

7. PLOS authors have the option to publish the peer review history of their article (what does this mean?). If published, this will include your full peer review and any attached files.

Reviewer #1: No

---

## [Editor Report · Acceptance letter]

22 Apr 2020

PONE-D-19-30981R1 

Seroprevalence of three paramyxoviruses; Hendra virus, Tioman virus, Cedar virus and a rhabdovirus, Australian bat lyssavirus, in a range expanding fruit bat, the grey-headed flying fox (Pteropus poliocephalus). 

Dear Dr. Boardman:

I am pleased to inform you that your manuscript has been deemed suitable for publication in PLOS ONE. Congratulations! Your manuscript is now with our production department. 

With kind regards,

on behalf of

Prof Wanda Markotter 

Academic Editor

PLOS ONE